Role of innate and acquired resilience in behavioral system, mental health, and internet addiction among Japanese adolescents in the COVID-19 pandemic

Kubo Takahiro songyou312@gmail.com 1
Masuyama Akihiro 2
Sugawara Daichi 3
1 Human Sciences, Tsukuba University , Bunkyo-ku , Tokyo , Japan
2 Psychology, Iryo Sosei University , Iwaki , Fukushima , Japan
3 Human Sciences, Tsukuba University , Tsukuba , Ibaraki , Japan
Greco Gianpiero
Electronic publication date: 2023 Feb 3
Publication date: 2023
Volume: 11
Electronic Location ID: e14643
Received 2022 Sep 15; Accepted 2022 Dec 6
Copyright: ©2023 Kubo et al.
Copyright year: 2023
Copyright holder: Kubo et al.
License: This is an open access article distributed under the terms of the Creative Commons Attribution License, which permits unrestricted use, distribution, reproduction and adaptation in any medium and for any purpose provided that it is properly attributed. For attribution, the original author(s), title, publication source (PeerJ) and either DOI or URL of the article must be cited.
License URL: https://creativecommons.org/licenses/by/4.0/

Keywords: Resilience, Depression, Behavioral system, Internet addiction

Funding: Japan Internet Safety Promotion Association (JISPA) This study was supported by the Japan Internet Safety Promotion Association (JISPA) under the “FY2020 Research Support (no grant number)”. There was no additional external funding received for this study. The funders had no role in study design, data collection and analysis, decision to publish, or preparation of the manuscript.

==============================
Background

This study examines mediation models in which behavioral inhibition and activation systems (BIS/BAS) impact internet addiction through mental health and the moderating roles of innate and acquired resilience in the models.

Methods

The data set used in this study was a cross-sectional survey among 952 adolescents in July 2021. Internet Addiction Test, Fear of COVID-19 Scale, BIS/BAS scales, and Depression Self-Rating Scale were used for analysis. After controlling for gender, the mediation and moderated mediation models were examined.

Results

The results revealed that depressive symptoms partially mediated the relationship between BIS and internet addiction and between BAS-fun-seeking (BAS-FS) and internet addiction. Innate and acquired resilience moderated the relationship between depressive symptoms and internet addiction. The indirect effect of innate and acquired resilience on internet addiction via depressive symptoms was statistically significant in both low and high innate and acquired resilience. The results of conditional indirect effect analysis indicated that the depressive symptoms–internet addiction association decreased with the increase of innate or acquired resilience level.

Discussion

Our results suggested that depression symptoms played a significant mediation role in the relationships between BIS/BAS and internet addiction, and higher innate and acquired resilience was associated with a reduced risk of internet addiction. BIS/BAS may be a risk for internet dependence via mental health, and innate and acquired resilience appears to serve as a protective factor.

Introduction

The spread of coronavirus disease 2019 (COVID-19) has dramatically changed our lives. Workplaces, educational settings, and family gatherings have been restricted to prevent the spread of infection. Our leisure time has also been transformed. The prevalence of individuals who indulged in physical activities decreased after the COVID-19 quarantine (Gjaka et al., 2021; Mauro et al., 2022). We are encouraged to use the internet for various activities. However, there is a concern about increasing problematic internet use, especially among the youth. Identifying the processes by which at-risk personalities develop internet addiction and discovering factors that prevent it will provide substantial evidence to alleviate internet addiction among the youth. Therefore, this study examines how an individual’s reward sensitivity and inhibitory function influence internet addiction via mental health and whether innate and acquired resilience might have protective effects.

Internet addiction

Although internet use provides various benefits in daily life, its excessive use can lead to internet addiction. Internet addiction or problematic internet use is characterized by excessive or poorly controlled preoccupations, desires, and behaviors related to internet use that lead to impairment and distress (Weinstein et al., 2014). The internet has a particularly significant impact on young people (Shotton, 1991). Excessive internet use can negatively impact schooling, family interaction, psychological well-being, and physical health (Young & De Abreu, 2011). The results of meta-analyses indicated that internet addiction is significantly associated with alcohol abuse, attention deficit hyperactivity, depression, and anxiety (Ho et al., 2014). During the pandemic, school-aged children’s increased time spent on social media and smartphones was associated with greater psychological distress (Chen et al., 2021). Fear of COVID-19 as a response to environmental stressors was associated with internet addiction disorder in Italian students (Servidio et al., 2021). Thus, internet addiction seems related to mental health problems and pandemic-specific stress.

BIS/BAS and internet addiction

Internet addiction is associated with abnormalities in an individual’s reward sensitivity, inhibitory function, and impulse regulation (Dong & Potenza, 2014; Brand et al., 2016). Addiction models frequently conceptualize reward and inhibition according to two primary brain and behavioral systems responding to punishing and reinforcing stimuli: the behavioral inhibition system (BIS) and the behavioral activation system (BAS) (Carver & White, 1994). According to Gray’s neuropsychological reinforcement sensitivity theory of personality (RST) (Gray, 1987), the BIS responds to punishment stimuli or reward termination. It evokes fear (negative affect) and withdrawal or avoidance behavior. The BAS responds to reward or cessation of punishment stimuli, producing positive affect and approach behaviors. BAS consists of tapping into vigorous and quick goal pursuit (Drive; BAS-D), receptivity to reward (Reward Responsiveness; BAS-R), and desire for new and potentially rewarding experiences (Fun Seeking; BAS-FS) (Carver & White, 1994).

Adolescents with higher BAS and BAS-FS showed a higher risk of internet addiction (Yen et al., 2012). BIS and BAS-FS scores were reportedly higher in students with internet addiction than in alcoholism (Yen et al., 2009). BIS has directly predicted positive internet addiction in girls, while BAS directly predicted positive internet addiction in boys (Li et al., 2019). BIS is not directly or indirectly associated with internet addiction, but BAS is associated with internet addiction only through depression and social anxiety in adults (Fayazi & Hasani, 2017). Thus, although BIS/BAS contributes to internet addiction among youth, it is inconsistent, and little is known about the mediating and regulating mechanisms underlying these associations. A study reported that anxiety and depression mediate BIS between internet addiction, while anxiety and impulsivity mediate BAS-FS (Park et al., 2013). Furthermore, BIS predicts fear of infection as a stressor specific to the current pandemic (Oniszczenko, 2021), and fear of disease has also been shown to influence internet addiction (Servidio et al., 2021). This suggests a possible association between BIS and BAS with internet addiction, mediated by poor mental health.

Moderating functions of resilience

Resilience is the ability to overcome the adverse effects associated with risk exposure. It helps individuals cope with traumatic experiences (Fergus & Zimmerman, 2005). It is an intrinsic capability and competence that can be acquired and developed (Alvord & Grados, 2005). A two-dimensional resilience scale (BRS) separately evaluates intrinsic factors (optimism, control, sociability, and vitality) and acquired resilience factors (problem-solving, self-understanding, and understanding others) (Hirano, 2010). In addition, both types of resilience have temporal stability (Ueno, Hirano & Oshio, 2020). Research has shown that innate and acquired resilience serve different psychological functions (Hirano, 2012; Ueno & Oshio, 2017).

Three models of resilience (compensatory, protective, and challenging) have been proposed (Fergus & Zimmerman, 2005). The protective model describes how specific factors moderate or reduce the impact of risk on adverse outcomes. Prior research has reported that the relationship between personality traits and depressive symptoms is moderated by resilience (Gong et al., 2020). Resilience was also shown to moderate the association between anxiety, impulsivity, and internet addiction (Nam et al., 2018). Thus, it is assumed that resilience moderates the association between personality traits and mental health, and the association between mental health and internet addiction, respectively. As resilience is characterized by high levels of positive emotions (Tugade & Fredrickson, 2004), the experiences of positive emotions might broaden an individual’s instantaneous thought–action repertories and build lasting personal resources such as physical, intellectual, and social resources to manage future threats (Fredrickson, 2001). Therefore, our study provides new insights into successful internet use among youth during and after the pandemic by examining whether two-factor resilience can reduce adverse effects associated with risk factors such as internet addiction and mental health.

The present study

This study first examined mediation models in which BIS and BAS influence internet addiction through mental health. The mental health variables assessed in this study were depression and fear of COVID-19. We hypothesized that BIS is fully mediated by mental health variables concerning internet addiction, while BAS is partially mediated. Second, we examined the moderating functions of innate and acquired resilience in the moderated mediation model (Fig. 1). Our study assumed that innate and acquired resilience moderates mental health and prevents internet addiction.

Figure 1 Hypothesized conceptual model of the moderated mediation.

BIS, behavioral inhibition system; BAS, Behavioral activation system.

Material and Methods

Study design and data collection

The current study is a cross-sectional design, which used the 2021 dataset of the “I’M HAPPY” project in Japan (Masuyama et al., 2021; https://osf.io/4dfb8/). A study with a different research perspective from the present study, focusing on depressive symptoms, has already been reported using this data set (Masuyama, Shinkawa & Kubo, 2020). Data collection for the project was performed in July 2021 at two junior high schools in Japan. The survey was conducted in cooperation with the boards of education and school principals of the two schools during the designated classes. The dataset included 965 adolescents. Three of them were excluded for missing values, and we finally analyzed 962 adolescents (469 girls, 48.8%; 493 boys, 51.2%). The first grade had 307 students aged 12–13 years, the second grade had 318 students aged 13–14 years, and the third grade had 337 students aged 14–15 years. The current study used measures of depressive symptoms, BIS/BAS, internet addiction, fear of COVID-19, and resilience. Informed consent was obtained from the board of education, school, and teachers. Informed consent from participants was obtained by providing their responses to the questionnaire. The study design and procedures comply with the principles of the Declaration of Helsinki. The overall procedure for this cross-sectional survey was approved by the ethics committee of Iryo Sosei University (Receipt number: 21-09).

Measures

Internet addiction test

We used the Japanese version of Internet Addiction Test (IAT) is a 20-item measure of the severity of internet addiction. The Japanese version of IAT has been used in other studies (Tateno et al., 2019; Inoue et al., 2021). The items were translated into Japanese by the National Hospital Organization Kurihama Medical and Addiction Center (n.d.) with permission from the author of the original scale (Young, 1998). Participants responded to each item on a five-point scale ranging from 1 (does not apply) to 5 (always applies). A higher score indicated greater severity of internet addiction. All items showed high internal consistency in this study (α = .917).

Fear of COVID-19 scale

We used the Japanese version of the Fear of COVID-19 Scale (FCV-19S) (Ahorsu et al., 2020) to assess the fear of COVID-19 among people. This scale was a seven-items measure. This version was validated by Masuyama, Shinkawa & Kubo (2020) and has reported sufficient reliability. The participants responded to each item on a four-point scale ranging from 1 (strongly disagree) to 4 (strongly agree). A higher score indicated a greater fear of COVID-19. In this study, total scores were used for the analysis. All items showed high internal consistency in this study (α = .843).

BIS/BAS

The sensitivity of BIS and BAS was indexed by the child version of Carver and White’s (1994) BIS/BAS scales (Muris et al., 2005), validated in Japan by Koseki et al. (2018), with sufficient reported reliability. Twenty items were scored on a four-point scale ranging from 0 (not true) to 3 (very true). It included the BIS, BAS reward responsiveness (BAS-RR), BAS fun seeking (BAS-FS), and BAS drive (BAS-D). The internal consistency of the BIS (α = .706), BAS-RR (α = .846), BAS-FS (α = .758), and BAS-D (α = .831) in the present study were good.

Depression self-rating scale

We used the Japanese version of The Depression Self-Rating Scale (DSRS) is an 18-item measure of depressive symptoms for children. The items were translated into Japanese by the Murata et al. (1996) with permission from the author of the original scale (Birleson, 1981). The DSRS has a single-factor structure, and each item is rated on a three-point scale ranging from 1 (never) to 3 (always). The higher the score, the greater the depressive symptoms. All items showed high internal consistency in this study (α = .826).

Bidimensional resilience scale

The Bidimensional Resilience Scale (Hirano, 2010) was developed and validated in Japan. This test is a 21-item measure with a five-point scale ranging from 1 (strongly disagree) to 5 (strongly agree). The scale measures two aspects of resilience: innate resilience, measured by 12 items, and acquired resilience, measured by nine items. The higher the score, the greater the resilience. The internal consistency of innate resilience (α = .896) and acquired resilience (α = .794) were high.

Data analysis

Data analyses were performed using IBM SPSS Statistics 23 (IBM SPSS Statistics for Windows, Version 23.0; SPSS, Inc., Chicago, IL, USA). The mediation and moderated mediation models were examined using the SPSS PROCESS macro 4.5 software (Hayes, 2017) to test our theoretical model. We used Model 4 in PROCESS to check the mediation model, including a bootstrapping procedure (5,000 bootstrap samples). The mediation effect was significant if the 95% confidence interval (CI) did not include 0 (Preacher & Hayes, 2008). The moderated mediation model was tested with Model 58 and was applied in the conceptual model shown in Fig. 1. This study examined a moderated mediation model to better understand the relationship between BIS/BAS and internet addiction. This model included mental health variables as mediators and innate and acquired resilience as moderator variables. A conditional indirect effect was assessed using the pick-a-point approach. Two innate and acquired resilience values, low innate and acquired resilience (M - SD), and high innate and acquired resilience (M + SD), were defined to test significant moderating effects. We included gender types as a covariate in the two models.

Results

Descriptive and correlational analyses

The descriptions and correlations of all the variables are presented in Tables 1 and 2. BIS and BAS-FS were positively associated with internet addiction, but BAS-D and BAS-FS were not. Depressive symptoms were positively associated with BIS and internet addiction and negatively associated with BAS-FS. Fear of COVID-19 was not associated with internet addiction. Innate resilience was negatively related to BIS and depressive symptoms and positively related to BAS-FS. Acquired resilience was positively associated with BIS, BAS-D, BAS-R, and BAS-FS and negatively associated with depressive symptoms.

Mediation model

The mediation model of BIS is depicted in Fig. 2A. BIS positively predicted depressive symptoms (B = .177, p < .001, 95% CI [.113–.241]) and internet addiction (B = .078, p = .015, 95% CI [.015–.140]). Depressive symptoms positively predicted internet addiction (B = .352, p < .001, 95% CI [.291–.413]). An indirect effect of BIS on internet addiction through depressive symptoms was significant (the indirect effect = .062, 95% CI [.034–.093]), which indicates a partial mediation of depressive symptoms.

The mediation model of BAS-FS is depicted in Fig. 2B. BAS-FS negatively predicted depressive symptoms (B =  − .167, p < .001, 95% CI [−.227–−.106]) and internet addiction (B = .157, p < .001, 95% CI [.098–.216]). Depressive symptoms positively predicted internet addiction (B = .393, p < .001, 95% CI [.332–.453]). An indirect effect of BAS-FS on internet addiction through depressive symptoms was significant (indirect effect = −.065, 95% CI [−.096–−.036]), which indicated a partial mediation of depressive symptoms.

Table 1 Descriptive statistics.

	M	SD	
BIS	18.14	4.21	
BAS_D	10.92	3.05	
BAS_R	15.05	3.6	
BAS-FS	10.4	2.78	
Depressive symptoms	11.21	6.25	
Trait anxiety	36.42	8.62	
Fear of COVID-19	17.29	6.2	
Innate resilience	39.38	9.9	
Acquired resilience	31.03	6.43	
Internet addiction	39.77	14.56	
Notes.

M mean

SD standard deviation

BIS behavioral inhibition system

BAS-D behavioral approach system (drive)

BAS-R behavioral approach system (reward responsiveness)

BAS-FS behavioral approach system (fun seeking)

COVID-19 coronavirus disease 2019

Table 2 Correlation coefficient.

	BIS	BAS_D	BAS_R	BAS-FS	Depressive symptoms	Trait anxiety	Fear of COVID-19	Innate resilience	Acquired resilience	
BIS										
BAS_D	0.2**									
BAS_R	0.3**	0.54**								
BAS-FS	0.2**	0.43**	0.6**							
Depressive symptoms	0.2**	−0.25**	−0.27**	−0.16**						
Trait anxiety	0.5**	−0.08*	0.01	0.06*	0.67**					
Fear of COVID-19	0.3**	0.04	0.12**	0.08*	−0.01	0.19**				
Innate resilience	−0.1*	0.32**	0.36**	0.31**	−0.61**	−0.44**	0.05			
Acquired resilience	0.1**	0.36**	0.42**	0.3**	−0.4**	−0.2**	0.08*	0.66**		
Internet addiction	0.2**	−0.02	0.01	0.09**	0.36**	0.37**	0.05	−0.32**	−0.24**	
Notes.

* p < .05.

** p < .01.

BIS behavioral inhibition system

BAS-D behavioral approach system (drive)

BAS-R behavioral approach system (reward responsiveness)

BAS-FS behavioral approach system (fun seeking)

COVID-19 coronavirus disease 2019

Figure 2 (A–B) Path coefficients for the mediation model.

Two asterisks (**) indicate indicates statistically significant (p < .01); one asterisk (*) indicates statistically significant (p < .05); BIS: behavioral inhibition system; BAS-FS, Behavioral activation system (fun-seeking); DS, depressive symptoms.

Moderated mediation model

Figures 3A and 3B show the results of the moderated mediation analyses for BIS. Innate resilience negatively predicted depressive symptoms (B =  − .587, p < .001, 95% CI [−.636–−.538]) and internet addiction (B =  − .156, p < .001, 95% CI [−.230–−.083]). Although the interaction of innate resilience and BIS did not have a significant effect on depressive symptoms (B =  − .001, p = .957, 95% CI [−.041–.039]), the interaction of innate resilience and depressive symptoms had a significant effect on internet addiction (B =  − .073, p = .005, 95% CI [−.124–−.023]). Acquired resilience negatively predicted depressive symptoms (B =  − .421, p < .001, 95% CI [−.477–−.365]) and internet addiction (B =  − .130, p < .001, 95% CI [−.195–−.064]). Although the interaction of acquired resilience and BIS did not have a significant effect on depressive symptoms (B = .016, p = .501, 95% CI [−.031–.062]), the interaction of acquired resilience and depressive symptoms had a significant effect on internet addiction (B =  − .086, p = .003, 95% CI [−.142–−.030]).

Figure 3 (A–D) Path coefficients for the moderated mediation model.

Two asterisks (**) indicate statistical significance (p < .01); one asterisk (*) indicates statistical significance (p < .05); BIS, behavioral inhibition system; BAS-FS, Behavioral activation system (fun-seeking); DS, depressive symptoms; IR, innate resilience; AR, acquired resilience.

Figures 3C and 3D present the results of the moderated mediation analyses for BAS-FS. Innate resilience negatively predicted depressive symptoms (B =  − .600, p < .001, 95% CI [−.652–−.548]) and internet addiction (B =  − .219, p < .001, 95% CI [−.294–−.144]). Besides, the interaction of innate resilience and BASFS did not have a significant effect on depressive symptoms (B =  − .019, p = .368, 95% CI [−.022, .059]), but the interaction of innate resilience and depressive symptoms had a significant effect on internet addiction (B =  − .070, p = .006, 95% CI [−.120–−.021]). Acquired resilience negatively predicted depressive symptoms (B =  − .384, p < .001, 95% CI [−.443–−.325]) and internet addiction (B =  − .159, p < .001, 95% CI [−.224–−.094]). Besides, the interaction of acquired resilience and BASFS did not have a significant effect on depressive symptoms (B = .002, p = .925, 95% CI [−.045–.049]), but the interaction of acquired resilience and depressive symptoms had a significant effect on internet addiction (B =  − .077, p = .006, 95% CI [−.132–−.022]). Interestingly, although BAS-FS had a significant effect on depression symptoms(B = .157, 95% CI [.098–.216]) in the previous mediation model (Fig. 2), BAS-FS did not have a significant effect on depression symptoms in the moderated mediation model (for innate resilience, B = .025, p = .3422, 95% CI [−.027–.077]; for acquired resilience, B =  − .053, p = .079, 95% CI [−.112–.006]).

To further study this moderation, the conditional indirect effect of two types of resilience and depressive symptoms on internet addiction was examined using the pick-a-point approach at low innate and acquired resilience (M − SD) and high innate and acquired resilience (M + SD). The indirect effect of innate resilience on internet addiction via depressive symptoms was statistically significant in both low innate resilience (effect = .326, p < .001, 95% CI [.240–.412]) and high innate resilience (effect = .194, p < .001, 95% CI [.098–.290]). The indirect effect of acquired resilience on internet addiction through depressive symptoms was statistically significant for low acquired resilience (effect = .395, p < .001, 95% CI [.310–.481]) and high acquired resilience (effect = .237, p < .001, 95% CI [.150–.325]). However, in the results shown in Fig. 4, the slope of the straight line reflecting the conditional effects of high and low resiliencies are different; the smaller slope of the straight line implies a greater moderating influence of resilience. When innate or acquired resilience was high, the slope was smaller than when they were low. Therefore, when students have high innate or acquired resilience, the impact of depression on internet addiction is reduced further.

Figure 4 The moderation of resilience of indirect effect of depression on internet addiction.

Note: DS, depressive symptoms; IR, innate resilience; AR, acquired resilience.

Discussion

This study examined a moderated mediation model to better understand the relationship between BIS/BAS and internet addiction. This model included mental health variables as mediators and innate and acquired resilience as moderator variables. As expected, the BAS-FS partially mediated the relationship between internet addiction and depressive symptoms. However, the BIS partially mediated the relationship between internet addiction and depressive symptoms, and no other BAS variables were associated with internet addiction. In addition, innate and acquired resilience moderated the association of depressive symptoms with internet addiction.

The mediation analysis showed that depressive symptoms partially mediated the impact of BIS and BAS-FS on internet addiction, which was consistent with previous studies (Park et al., 2013; Nam et al., 2018). However, the present study showed that BIS positively predicted depression and internet addiction, while BAS-FS negatively predicted depression and positively predicted depression. In other words, BAS-FS suppresses internet addiction by improving mental health but directly enhances internet addiction. As BAS-FS indicates a disposition to pursue enjoyment, a high BAS-FS may be both promotion of positive experiences and addictive behaviors. In support of this, the path from BAS-FS to depression became non-significant when the resilience variable, which is closely related to increased positive experiences (Tugade & Fredrickson, 2004), was inputted in the model. However, previous studies have shown that BAS-FS positively predicts depression (Park et al., 2013; Nam et al., 2018). Hence, how people engage in activities when the BAS-FS is activated must be evaluated. Research on the passion for online gaming or smartphone usage has shown that harmonious passion does not predict addiction or problematic usage, but obsessive passion does (Wang & Chu, 2007; Kubo & Sawamiya, 2021). Future research should include additional variables that influence engagement with the internet, such as passion.

The moderated mediation showed that innate and acquired resilience not only exerted a direct effect on depressive symptoms and internet addiction but also moderated the association of depressive symptoms with internet addiction. The results of conditional indirect effect analysis indicated that the depressive symptoms–internet addiction association decreased with the increase of resilience level. Importantly, this effect equally well whether innate or acquired resilience. Thus, these higher levels of resilience can be expected to prevent the induction of internet addiction caused by deteriorating mental health. This is also consistent with a previous study that showed that resilience moderated the association between anxiety and impulsivity and internet addiction (Nam et al., 2018). Advancing and preparing for resilience before exposure to risks of mental health decline, such as pandemics, may indirectly help prevent internet addiction.

Several results were identified that differed from expectations. The present study found that fear of COVID-19 was not a major factor in increasing internet addiction among youth. This result was inconsistent with previous studies (Servidio et al., 2021). In this regard, as fear of COVID-19 has been found to be associated with boredom and reduced physical activity (Bösselmann et al., 2021), the association between fear of COVID-19 and internet addiction may become more apparent when these variables are introduced as mediating variables.

When preventing or intervening in internet addiction, considering the characteristics of BIS and BAS could lead to more effective approaches in adolescents. As some resilience is innate and some can also be acquired through learning, it can be assumed that implementing a school-based resilience enhancement program would be effective not only in improving mental health but also in preventing internet addiction. However, as individuals with high BAS tend to exhibit risky behaviors and internet addiction mediated by impulsivity (Braddock et al., 2011; Park et al., 2013), it seems necessary to examine the role of innate and acquired resilience for impulsivity as well.

Limitations

A few limitations of this study should be noted. First, this study was conducted exclusively with a sample of Japanese adolescents and, in turn, only two junior high schools; thus, it is not representative of the Japanese population. The sample also included those with internet addiction and those without addiction. It will be essential to survey Japanese people of other age groups such as a sample of young people addicted to the internet to examine the model of this study. Second, it was a cross-sectional study. Further research using a longitudinal design is needed to strengthen the present findings. Third, this study relied solely on self-reported measurements, which could have induced bias. Future studies should measure resilience objectively, such as through reports from teachers.

Conclusion

In summary, BIS and BAS-FS partially mediated the association between depression and internet addiction. Notably, innate and acquired resilience moderated the relationship between depressive symptoms and internet addiction. In other words, higher innate and acquired resilience was associated with a reduced risk of internet addiction. Thus, the two cerebral and behavioral systems that respond to punitive and reinforcing stimuli are risk factors for internet addiction in youth, and resilience is a protective factor.

Additional Information and Declarations

Competing Interests

Author Contributions

Human Ethics

Data Availability

The authors declare there are no competing interests.

Takahiro Kubo conceived and designed the experiments, performed the experiments, analyzed the data, prepared figures and/or tables, authored or reviewed drafts of the article, and approved the final draft.

Akihiro Masuyama performed the experiments, authored or reviewed drafts of the article, and approved the final draft.

Daichi Sugawara conceived and designed the experiments, authored or reviewed drafts of the article, and approved the final draft.

The following information was supplied relating to ethical approvals (i.e., approving body and any reference numbers):

The overall procedure for this longitudinal survey was approved by the ethics committee of Iryo Sosei University (Receipt number: 21-09).

The following information was supplied regarding data availability:

The data is available at OSF: Masuyama, Akihiro, Takahiro Kubo, Hiroki Shinkawa, Daichi Sugawara, and Ayana Noto. 2022. “I’M HAPPY Project.” OSF. April 29. doi: 10.17605/OSF.IO/4DFB8.

The variables used from “21ss_raw” were Gender (3), DSRS (4-21), FCV−19 V (47-53), IAT (54-73), Bidimensional Resilience (82-102) and BIS/BAS (103-122). The second column of the Excel sheet also shows the correspondence between each variable and the numeric category.

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
