# Peer review of "Role of innate and acquired resilience in behavioral system, mental health, and internet addiction among Japanese adolescents in the COVID-19 pandemic"

_PeerJ, doi:10.7717/peerj.14643_

## Round 0.1 · original submission · Minor Revisions

Dear Authors,

Please reply point by point to the reviewers' comments and resubmit.

Reviewer 1 ·

Basic reporting

It is interesting research that contributes to understanding the role of influence Internet addiction through mental health with the behavioral inhibition system (BIS) and the behavioral activation system (BAS) during the time of COVID-19 pandemic.

The paper has been written well. It is very important that authors had obteined an ethical approval. The paper is appropriate, the sample is adequate and it has current references. The authors confirm the good psychometric properties of all questionnaires used. Data analysis is correct for the objective of this study.

I have minor revision. I have reported my comments in the following:
- DISCUSSION: However, since previous studies have shown that BAS-FS positively predicts depression (Park et al., 2013; Nam et al., 2018), a more detailed examination is also warranted. Please add more information about how in the future new research could investigate this point

- the lack of an association between fear of COVID-19 and Internet addiction may be due to the involvement of multiple factors related to fear of COVID-19, including COVID-19 infection status and school situations. Please add more references to this idea

Experimental design

It is ok.


The paper has been written well. It is very important that authors had obteined an ethical approval. The paper is appropriate, the sample is adequate and it has current references. The authors confirm the good psychometric properties of all questionnaires used. Data analysis is correct for the objective of this study.

Validity of the findings

These are ok



The paper has been written well. It is very important that authors had obteined an ethical approval. The paper is appropriate, the sample is adequate and it has current references. The authors confirm the good psychometric properties of all questionnaires used. Data analysis is correct for the objective of this study.

Additional comments

I have not any additional comments

Accept manuscript with minor revisions.

·

Basic reporting

TITLE
I suggest changing the title to specify that they are Japanese adolescents. The title could be "Role of innate and acquired resilience in behavioral system, mental health, and Internet addiction among Japanese adolescents in the COVID-19 pandemic".

INTRODUCTION
Although the introduction is well written and comprehensive, it does not include all relevant references to support the background provided. The first reference even appears in line 56 and from that point on many of these are quite dated (For example, Shotton, 1991; Carver & White, 1994).

(line 44): “Our leisure time has also been transformed.” Please consider the following references: “Mauro, M., et al. (2022). Effects of quarantine on Physical Activity prevalence in Italian Adults: a pilot study. PeerJ, 10, e14123” ; “Gjaka, M., et al. (2021). The effect of COVID-19 lockdown measures on physical activity levels and sedentary behaviour in a relatively young population living in Kosovo. Journal of clinical medicine, 10(4), 763”.

Experimental design

METHODS
Please report study design.

The methods should be described more clearly. In fact, authors stated that they used the 2021 dataset of the I'M HAPPY project in Japan (Masuyama et al., 2021) and then they described the data collection of that project. However, below they reported: "We obtained informed verbal consent..." Why authors reported that they obtained some information if they used a dataset? Even if some authors are the same, informed verbal consents were obtained during that project and not for this study. It is just a note to improve the quality of the writing style.

Furthermore, the reference Masuyama et al., 2021 does not specify the journal in which the paper was published and for this reason I was not able to find the information of the project. Hence, please provide a complete citation and moreover, I suggest to also include in this paper some details concerning the procedure as data collection setting.

Some details of the Internet Addiction Test (IAT) should be reported. For example, the number of the items should be provide.

The above comment also for the Fear of COVID-19 Scale and the Depression Self-Rating Scale.

Of all the scales used the authors rightly report the internal consistency index. However, it is necessary to specify if they themselves calculated it for each scale or if they report the values of other authors. In the latter case, authors should add related references.

Validity of the findings

DISCUSSION
All results found should be argued more thoroughly.

Was the hypothesis formulated in the introduction by the authors confirmed or not?

Among the limitations of the study, authors should report that they investigated only Japanese adolescents and, in turn, only two junior high schools and that this study cannot be representative of the Japanese population.

Additional comments

None.

·

Basic reporting

Figure 3: I suggest using different arrows to indicate the discriminate principal mediation model (BIS-DS-IA) and the moderators (IR - IRX...).

Table 1: if possible, I suggest making different tables for mean values and correlations (or add a separator line)

Experimental design

The experimental design is well organized, but some information needs to be clarified.

Materials and Metods:
The authors not reported information on the inclusion and exclusion criteria. Please, add this information in the text.

Authors stated that they used a Fear of COVID-19 Scale validated by Masuyama, Shinkawa, and Kubo in 2022. How did they use ta scale that was introduced in the 2022, in a 2021 data collection?

Line 143: it is not a longitudinal study. Please, correct it

Statistical analysis:
Could the authors better specify the models reported in this paragraph? (as done in lines 271-272)

Validity of the findings

The article is interesting, and the results well presented and discussed

Additional comments

lines 60 - 61: the meaning of the sentence is not clear. Please, correct it

---

## Round 0.2 · accepted · Accept

Dear authors,

You have successfully addressed the reviewers' comments and the manuscript is now much improved.